# Anesthetic- and Analgesic-Related Drugs Modulating Both Voltage-Gated Na^+^ and TRP Channels

**DOI:** 10.3390/biom14121619

**Published:** 2024-12-18

**Authors:** Eiichi Kumamoto

**Affiliations:** Faculty of Medicine, Saga University, 5-1-1 Nabeshima, Saga 849-8501, Japan; kumamote@cc.saga-u.ac.jp

**Keywords:** TRP channel, voltage-gated Na^+^ channel, action potential, nerve conduction, excitatory synaptic transmission, L-glutamate, spinal cord, brain stem, lamina II, antinociception

## Abstract

Nociceptive information is transmitted by action potentials (APs) through primary afferent neurons from the periphery to the central nervous system. Voltage-gated Na^+^ channels are involved in this AP production, while transient receptor potential (TRP) channels, which are non-selective cation channels, are involved in receiving and transmitting nociceptive stimuli in the peripheral and central terminals of the primary afferent neurons. Peripheral terminal TRP vanilloid-1 (TRPV1), ankylin-1 (TRPA1) and melastatin-8 (TRPM8) activation produces APs, while central terminal TRP activation enhances the spontaneous release of L-glutamate from the terminal to spinal cord and brain stem lamina II neurons that play a pivotal role in modulating nociceptive transmission. There is much evidence demonstrating that chemical compounds involved in Na^+^ channel (or nerve AP conduction) inhibition modify TRP channel functions. Among these compounds are local anesthetics, anti-epileptics, α_2_-adrenoceptor agonists, antidepressants (all of which are used as analgesic adjuvants), general anesthetics, opioids, non-steroidal anti-inflammatory drugs and plant-derived compounds, many of which are involved in antinociception. This review mentions the modulation of Na^+^ channels and TRP channels including TRPV1, TRPA1 and TRPM8, both of which modulations are produced by pain-related compounds.

## 1. Introduction

Noxious stimuli delivered to the periphery are conveyed to the central nervous system (CNS) by action potentials (APs) that travel along primary afferent neuron fibers (see [1] for review). Primary afferent neurons express transient receptor potential (TRP) channels, i.e., non-selective cation channels, that are involved in transmitting sensory information, such as nociception and temperature (see [2] for review). Among these channels are TRPV (vanilloid; 1–6) involved in pain and hot, TRPA (ankyrin; 1) involved in pain and nociceptive cold (<17 °C), TRPM (melastatin; 1–8) involved in pain and mild cold, and TRPC (canonical; 1–7) involved in pain (see [3,4,5] for review). In fact, the cell bodies of primary afferent [dorsal root ganglion (DRG) and trigeminal ganglion (TG)] neurons express mRNA and protein for TRPV1, TRPA1 and TRPM8 [6,7,8,9,10]. The other types of TRP channel, such as TRPM2, TRPM3, TRPC5 and TRPC6, are also located in primary afferent neurons [11,12,13,14].

TRP channels are involved not only in pain and temperature but also in mechano-sensation (TRPC3 and 6 [15]), itch (TRPA1, TRPV1, 3 and 4 [16]) and a variety of sensory allergic reactions (TRPA1 [17]). TRP channels synthesized in the cell bodies are conveyed by axonal transport to their peripheral and central terminals and are thus involved in transmitting sensory information. Peripheral terminal TRP activation is involved in the production of AP, while central terminal TRP activation results in enhanced spontaneous release of L-glutamate from the terminal on spinal cord and brain stem lamina II neurons (see [18,19,20] for TRPV1; see [21,22,23,24] for TRPA1; see [20,23,24,25,26,27] for TRPM8; see [28] for TRPC3/6; for review, see [3,5,29,30]). The enhanced spontaneous L-glutamate release is caused by an increase in intracellular Ca^2+^ concentration, which is due to activation of voltage-gated Ca^2+^ channels via membrane depolarization occurring as a result of opening of TRP, or by Ca^2+^ passing through the TRP itself [3]. The lamina II neurons play a pivotal role in regulating nociceptive transmission from the periphery [31,32,33]. Consistent with this idea, glutamatergic excitatory transmission in lamina II neurons is suppressed by various endogenous analgesics, including neuropeptides (for example, see [34,35,36]).

Regarding nociceptive transmission, in primary afferent neurons, which transmit information as APs involving Na^+^ channel activation, TRPs at their peripheral terminals receive nociceptive stimuli, while TRPs at their central terminals modify the transmission of nociceptive information. The involvement of TRP in nociceptive transmission is supported by (1) upregulated TRPV1 expression in the spinal lamina II following peripheral inflammation [37], (2) reversal by intrathecally-applied TRPA1 antagonist of mechanical and cold hyperalgesia (an increased sensitivity to pain) caused by the intra-plantar injection of complete Freund’s adjuvant [38], (3) chronic neuropathic pain (chronic pain lasting more than three months after nerve injury or dysfunction; see [39] for review) alleviation by TRPM8 activation in primary afferent neurons [40], (4) reduced heat-induced thermal hyperalgesia following inflammation in TRPM3-deficeint mouse models [12], (5) reduced nociceptive (thermal and mechanical) thresholds following inflammation in TRPC5-knockout mouse models [41], (6) the observation that knockdown of TRPM2 in the DRG and spinal cord alleviated neuropathic pain produced by sciatic nerve chronic constriction injury [42], and (7) analgesia produced by a TRPC6 inhibitor in rat neuropathic pain models induced by spared nerve injury [14].

The TRP channels are targets of various compounds derived from plants (see [17,43,44] for review). For instance, TRPV1, TRPA1 and TRPM8 are activated by capsaicin (8-methyl-*N*-vanillyl-6-nonenamide, the major pungent ingredient in hot peppers [45], having a vanillyl group), allyl isothiocyanate (AITC, the pungent principal in mustard oil or wasabi [46]) and menthol (2-isopropyl-5-methylcyclohexanol, a secondary alcohol in peppermint or other mint oils [47]), respectively (see [4] for review).

Although many of the properties of the TRP channels are examined in the cell body of primary afferent neuron, there seems to be a difference in activation by plant-derived chemicals between primary afferent cell body and central terminal TRP, as indicated by previous studies (see [29] for review). Such a difference may be due to interactions between TRP, such as TRPA1-TRPV1 complex [48], the presence of TRP splice variants [49,50], or TRP modulation by intracellular second messengers [51,52,53]. Many plant-derived compounds, which increased spontaneous L-glutamate release through central terminal TRP activation in lamina II neurons, inhibited nerve AP conduction and Na^+^ channels involved in AP production (see [54] for review). As with plant-derived compounds, both TRP and Na^+^ channels are modulated by local anesthetics, general anesthetics, anti-epileptics, opioids, adrenoceptor agonists, antidepressants and non-steroidal anti-inflammatory drugs (NSAIDs), all of which are involved in antinociception. In this review, how these drugs and the other pain-related drugs affect the primary afferent neuron, its central terminal, cloned TRP and Na^+^ channels (nerve conduction) is shown, and then a modulatory site on TRP is suggested to be similar in chemical structure to that of Na^+^ channels.

## 2. Local Anesthetics

Local anesthetics are used not only for AP conduction blocking (see [55,56] for review) in surgical anesthesia but also for chronic pain alleviation as analgesic adjuvant (see [57,58] for review). Leffler et al. [59,60] have reported that a typical local anesthetic lidocaine, which blocks Na^+^ channels (with a half-maximal inhibitory concentration (IC_50_) value of 0.204 mM in *Xenopus laevis* sciatic nerve fibers [61]), activates TRPV1 and TRPA1 (half-maximal effective concentration (EC_50_): 3.4 and 5.7 mM, respectively) expressed in rat DRG neurons. Consistent with this observation, lidocaine at concentrations higher than 2 mM activated primary afferent central terminal TRPA1, resulting in spontaneous L-glutamate release enhancement in rat spinal lamina II neurons (spontaneous excitatory postsynaptic current (sEPSC) frequency increase: 132% at 5 mM [62]). The primary afferent central terminal TRPA1 in the rat spinal lamina II was also activated by another local anesthetic tetracaine (307% sEPSC frequency increase at 5 mM [63]), which blocked Na^+^ channels in *Xenopus laevis* sciatic nerve fibers (IC_50_ = 0.0007 mM [61]). Lidocaine and tetracaine inhibited Na^+^ channel-mediated frog sciatic nerve compound action potentials (CAPs; IC_50_ for lidocaine and tetracaine: 0.74 mM [64] and 0.014 mM [65], respectively). Bupivacaine, which inhibited Na^+^ channels (at 0.3 mM in rat brain [66]; IC_50_ = 0.027 mM in *Xenopus laevis* sciatic nerve fibers [61]) and frog sciatic nerve CAPs (76% amplitude reduction at 0.5 mM [67]), activated TRPA1 in rat DRG neurons (EC_50_ = 0.41 mM [68]). CAP inhibition similar to that in the frog sciatic nerve has been also reported in mammalian nerve fibers. For example, lidocaine suppressed rat sciatic nerve CAPs with an IC_50_ value of 0.28 mM [69]. Tetracaine and bupivacaine depressed rabbit fast-conducting A fiber CAPs (IC_50_: 0.009 and 0.048 mM, respectively [70]).

A cationic lidocaine derivative QX-314 (*N*-ethyl-lidocaine), which inhibits Na^+^ channels when intracellularly applied, is known to have an ability to go through TRPV1 [71,72]. This indicates that QX-314 suppresses Na^+^ channels in primary afferent neurons expressing TRPV1. QX-314 exhibited a biphasic effect on human TRPV1 expressed in *Xenopus laevis* oocytes, activating at high concentrations (such as 30–60 mM) and inhibiting at low concentration (IC_50_ = 8 μM for capsaicin response [73]), showing that QX-314 has an effect on TRPV1. Moreover, QX-314 is shown to activate not only human TRPV1, but also TRPA1 expressed in human embryonic kidney (HEK) 293 cells [74]. The results of these local anesthetic actions are summarized in Table 1.

## 3. General Anesthetics

General anesthetics are thought to produce not only the loss of consciousness but also analgesia by acting on ligand-gated ion channels (see [75,76] for review), for example, in a pain inhibitory circuit of the amygdala [77]. A general anesthetic propofol (2,6-diisopropylphenol) activated rat TRPV1 [78] and TRPA1 (at 3–300 μM [79]) expressed in HEK293 cells, and also TRPV1 and TRPA1 located in rodent DRG neurons (at 10–30 μM [80,81]), while inhibiting Na^+^ channels (IC_50_ = 20 μM in the human brain [82]; at >60 μM in *Xenopus laevis* myelinated nerve fibers [83]; IC_50_ = 46 μM in rat cerebro-cortical synaptosomes [84]; IC_50_ = 10 μM for rat brain IIa Na^+^ channels expressed in Chinese hamster ovary (CHO) cells [85]; see [86] for review). Ton et al. [87] have reported that propofol has a modulatory action on rodent but not human TRPA1, indicating a species difference. Frog sciatic nerve CAPs were inhibited by propofol (IC_50_ = 0.14 mM); this inhibition required a specific chemical structure of propofol [88].

**Table 1 biomolecules-14-01619-t001:** Analgesic, analgesic adjuvants and other drugs exhibiting both TRP and Na^+^ channel (or CAP) modulation.

Chemical Compounds	Central Terminal TRP Modulation	Primary Afferent Neuron and Cloned TRP Modulation	Voltage-Gated Na^+^ Channel	CAP	References
**Local anesthetics**					
Lidocaine	TRPA1 ↑	TRPA1 ↑ TRPV1 ↑	↓	↓	[59,60,61,62,64,69]
Tetracaine	TRPA1 ↑	n.d.	↓	↓	[61,63,65,70]
Bupivacaine	n.d.	TRPA1 ↑	↓	↓	[61,66,67,68,70]
QX-314	n.d.	TRPV1 (↑, ↓) TRPA1 ↑	↓ (intracellularly-applied)	n.d.	[73,74]
**General anesthetics**					
Propofol	n.d.	TRPV1 ↑ TRPA1 ↑	↓	↓	[78,79,80,81,82,83,84,85,87,88]
Isoflurane	n.d.	TRPA1 (↑, ↓)	↓	n.d.	[79,87,89]
**Antiepileptics**					
Phenytoin	n.d.	TRPA1 ↑	↓	↓	[67,90,91,92]
Carbamazepine	n.d.	TRPV1 ↓	↓	↓	[67,93,94]
**Opioids**					
Morphine	n.d.	TRPV1 ↑ TRPA1 ↑	↓	↓	[95,96,97,98,99]
Tramadol	n.d.	TRPV1 (↑, →) TRPA1 ↓	↓	↓	[64,97,100,101,102,103]
*O*-Desmethyl tramadol	n.d.	TRPA1 ↓	n.d.	↓	[64,103]
**α_2_-Adrenoceptor agonist**					
Dexmedetomidine	n.d.	TRPV1 ↓ TRPM2 ↓	↓	↓	[65,104,105,106,107,108,109]
**Antidepressants**					
Duloxetine	n.d.	TRPV1 ↓ TRPM2 ↓ TRPC5 ↓	↓	↓	[110,111,112,113,114]
Maprotiline	n.d.	TRPM3 ↓	↓	↓	[113,115,116]
Amitriptyline	n.d.	TRPV1 ↓	↓	↓	[113,116,117,118,119,120,121]
Fluoxetine	n.d.	TRPV1 ↓	↓	↓	[113,116,122,123]
Desipramine	n.d.	TRPV1 ↓	↓	↓	[113,116,123,124]
**NSAIDs**					
Diclofenac	n.d.	TRPA1 ↑ TRPM3 ↓	↓	↓	[115,125,126,127,128]
Etodolac	n.d.	TRPA1 ↑	n.d.	↓	[126,129]
Tolfenamic acid	n.d.	TRPM3 ↓ TRPC6 ↓	↓	↓	[126,130,131]
Meclofenamic acid	n.d.	TRPM3 ↓ TRPC6 ↓	↓	↓	[126,130,131]
Mefenamic acid	n.d.	TRPM3 ↓ TRPC6 ↓	↓	↓	[126,130,131]
Flufenamic acid	n.d.	TRPM3 ↓ TRPC6 ↓	↓	↓	[126,130,131,132]
Ibuprofen	n.d.	TRPA1 ↑ (at 10 mM)	No effect at 1 mM	n.d.	[126,133]
Salicylate	n.d.	TRPM7 ↓	↓	n.d.	[134,135]
**Other drugs**					
Methylglyoxal	TRPV1 ↑ TRPA1 ↑	TRPA1 ↑	Nav1.8 ↑	n.d.	[136,137,138]
Nicotine	n.d.	TRPV1 ↑	↓	n.d.	[139]
Pregnenolone sulfate	n.d.	TRPV1 ↓ TRPM3 ↑	↓	n.d.	[115,140,141,142]
Riluzole	n.d.	TRPC5 ↑	↓	n.d.	[143,144]
Anandamide	TRPV1 ↑	TRPV1 ↑	↓	n.d.	[19,145,146,147]

n.d.: not determined; ↑: activation; ↓: inhibition; →: no change.

Another general anesthetic, isoflurane ((*RS*)-2-chloro-2-(difluoro-methoxy)-1,1,1-trifluoroethane), is shown to activate rat TRPA1 expressed in HEK293 cells (EC_50_ = 0.18 mM [79]) while having both rodent TRPA1 inhibition (IC_50_ = 0.28 mM for AITC response at a holding potential of −100 mV [87]) and Na^+^ channel inhibition in pyramidal neurons of the mouse prefrontal cortex (36% amplitude reduction at 0.45 mM at a holding potential of −70 mV [89]). The results of these general anesthetic actions are summarized in Table 1.

## 4. Antiepileptics

Antiepileptics are widely used as analgesic adjuvants to alleviate neuropathic and inflammatory pain (caused by inflammation leading to inflammatory cytokine release; see [148,149] for review). A typical anti-epileptic phenytoin (5,5-diphenylhydantoin) activated human TRPA1 but not TRPV1 and TRPM8 expressed in HEK cells (at 0.1 mM [90]), while inhibiting Na^+^ channels in rat cortical neurons (IC_50_ = 51 μM [91]) and N4TG1 mouse neuroblastoma cells (IC_50_ = 58 μM [92]), and frog sciatic nerve CAPs (16% amplitude reduction at 0.1 mM [67]). Carbamazepine (an iminostilbene derivative; 5H-dibenz[b,f]azepine-5-carboxamide) having an ability to suppress Na^+^ channels in mouse central neurons [93] and frog sciatic nerve CAPs (IC_50_ = 0.50 mM [67]) is suggested to inhibit rat TRPV1 from the observation that capsaicin-induced hyperalgesia was attenuated by carbamazepine [94]. TRPV1 is thought to be a potential target of anti-epileptics [150]. The results of these antiepileptic actions are summarized in Table 1.

## 5. Opioids

Opioids can exert analgesic effects not only in the CNS [see [151,152,153] for review] but also in the peripheral nervous system by activating opioid receptors (see [154,155] for review). TRPV1 and TRPA1 located in mouse DRG neurons were activated by morphine with an EC_50_ value of 2.6 mM [95]; morphine inhibited Na^+^ channels (frog Ranvier node [96]; IC_50_ = ca. 0.3 mM for TTX-sensitive currents in DRG neuroblastoma hybridoma cell line ND7/23 cells [97]) and frog sciatic nerve CAPs (15% amplitude reduction at 5 mM [98]). A similar AP conduction inhibition by morphine has been reported in the rat sciatic nerve; morphine reduced the peak amplitude of AP recorded intracellularly from DRG neurons with an IC_50_ value of 2.9 mM [99].

A clinically-used and orally-active μ-opioid receptor agonist tramadol ((1*RS*; 2*RS*)-2-[(dimethyl-amino) methyl]-1-(3-methoxyphenyl)-cyclohexanol hydrochloride [156]), which inhibited Na^+^ channels (IC_50_ = 0.103 mM for TTX-sensitive Nav1.2 [100]; IC_50_ = ca. 0.2 mM for TTX-sensitive channels in ND7/23 cells [97]) and frog sciatic nerve CAPs (IC_50_ = 2.3 mM [64]), activated rat TRPV1 expressed in CHO cells (EC_50_ = 0.08 μM [101]). An inhibitory action of tramadol on CAPs has been reported for the rat sciatic nerve [102]. Since tramadol (5 mg/kg) inhibits the spontaneous release of L-glutamate from nerve terminals by activating opioid receptors in the rat spinal cord in vivo ([157]; see [158,159] for EPSC suppression by μ-opioid receptor activation in the lamina II of tissue slice preparations), it is difficult to determine whether tramadol acts on primary afferent central terminal TRP. Miyano et al. [103] have reported that tramadol and its metabolite *O*-desmethyl tramadol (M1; at each 10 μM [160]) do not activate human TRPV1 and TRPA1 expressed in HEK293 cells while suppressing human TRPA1 (activated by AITC), but not human TRPV1 (activated by capsaicin), expressed in HEK293 cells. The difference between Marincsák et al. [101]’s and Miyano et al. [103]’s results on TRPV1 activation has been attributed to the distinction in cell line (CHO or HEK293) used to express TRPV1, rather than different species (human or rat [103]). Unlike tramadol, M1 exhibited a small reduction in frog sciatic nerve CAP peak amplitude (9% amplitude reduction at 5 mM [64]). The frog sciatic nerve CAP inhibitions produced by morphine and tramadol may be due to a membrane hyperpolarization occurring as a result of opioid-receptor activation. However, this is unlikely, because the morphine and tramadol activities were resistant to a non-selective opioid-receptor antagonist naloxone [64,98]. The results of these opioid actions are summarized in Table 1.

## 6. α2-Adrenoceptor Agonist

α_2_-Adrenoceptor agonists are widely used not only as an analgesic (which inhibits nociceptive transmission by activating α_2_ adrenoceptors in the CNS; for example, see [161,162]) but also as an analgesic adjuvant to alleviate chronic pain (see [163,164,165] for review). An α_2_-adrenoceptor agonist dexmedetomidine (DEX; ((+)-(*S*)-4-[1-(2,3-dimethylphenyl)-ethyl]-1H-imidazole; see [166,167] for review) inhibited Na^+^ channels in NG108-15 neuronal cells (22% amplitude reduction at 10 μM [104]) and also frog sciatic nerve CAPs (IC_50_ = 0.40 mM [65]). A similar CAP inhibitory action of DEX has been reported in the rat sciatic nerve [105]. On the other hand, TRPV1 activation by capsaicin in rat DRG neurons was suppressed by DEX (1 pM); this action has been attributed to inhibited adenylate cyclase/cyclic AMP/protein kinase A pathway following adrenoceptor activation [106]. DEX-induced TRPV1 inhibition has been also reported in mouse DRG neurons (30% capsaicin response suppression at 10 μM [107]). DEX inhibited not only TRPV1 (activated by capsaicin) but also TRPM2 (activated by its agonist cumene hydroperoxide) in mouse DRG and hippocampal neurons [108]. On the other hand, primary afferent central terminal TRPV1 in the rat spinal lamina II was unaffected by DEX at 0.3 μM [109]. This may have been due to the fact that this concentration is lower than those that inhibit Na^+^ channels in NG108-15 cells (see above; [104]) and frog sciatic nerve CAPs (76% amplitude reduction at 0.5 mM [65]). Alternatively, the lack of the inhibitory effect of DEX on central terminal TRPV1 may indicate that endogenous TRPV1 agonists such as anandamide (arachidonoylethanolamide; see [168,169] for review) do not constitutively promote spontaneous L-glutamate release in the lamina II. The results of the α_2_-adrenoceptor agonist actions are summarized in Table 1.

## 7. Antidepressants

Various types of antidepressants are used as analgesic adjuvants for treating neuropathic pain (see [170,171,172] for review). An antidepressant duloxetine (serotonin and noradrenaline reuptake inhibitor, SNRI) used for chronic pain alleviation inhibited both TRP (TRPV1 and TRPM2; activated by capsaicin and cumene hydroperoxide, respectively) in rat DRG neurons [110] and Na^+^ channels (IC_50_ = 22 μM for tetrodotoxin (TTX)-sensitive Nav1.7 expressed in HEK293 cells [111]; IC_50_ = 14 μM for TTX-resistant Nav1.5 [112]). Frog sciatic nerve CAPs have been shown to be inhibited by duloxetine (IC_50_ = 0.23 mM [113]). Moreover, duloxetine has been reported to inhibit TRPC5 that is involved in neuropathic and inflammatory pain (IC_50_ = 0.30 μM for a TRPC5 agonist riluzole (2-amino-6-[trifluoromethoxy]benzothiazole) response at negative potentials [114]; see [5] for review). On the other hand, another antidepressant, maprotiline (tetracyclic secondary amine), suppressed mouse TRPM3 expressed in HEK293 cells (IC_50_ = 1.3 μM for its agonist pregnenolone sulfate response [115]) while inhibiting Na^+^ channels (IC_50_ = 28 μM for human Nav1.7 expressed in HEK293 cells [116]) and frog sciatic nerve CAPs (IC_50_ = 0.95 mM [113]). Behavioral studies suggest that TRPV1 may be suppressed by other antidepressants, amitriptyline (tricyclic tertiary amine [117]), fluoxetine (selective serotonin reuptake inhibitor, SSRI [122]) and desipramine (tricyclic secondary amine [124]), all of which inhibited frog sciatic nerve CAPs (IC_50_ = 0.26, 1.5 and 1.6 mM, respectively, for amitriptyline, fluoxetine and desipramine [113]) and Na^+^ channels (amitriptyline: IC_50_ = 85 μM for human Nav1.7 expressed in HEK293 cells [116], IC_50_ = 20.2 μM in bovine adrenal chromaffin cells [118], the rat striatum [119], rat cortical neurons [120]; fluoxetine: IC_50_ = 74 μM for human Nav1.7 [116], IC_50_ = 1.11 μM at −60 mV in rat hippocampal neurons [123]; desipramine: IC_50_ = 24 μM for human Nav1.7 [116], IC_50_ = 1.68 μM at −60 mV in rat hippocampal neurons [123]). Sciatic nerve AP conduction suppression by antidepressants has been reported in mammalian nerve fibers. For instance, amitriptyline at 5 and 10 mM produced a complete nerve conduction blockade of rat sciatic nerve [121]. The results of these antidepressant actions are summarized in Table 1.

## 8. NSAIDs

Analgesia produced by NSAIDs has been attributed to various actions, including suppression of the synthesis of prostaglandins from arachidonic acid by inhibiting cyclooxygenase (see [173,174,175] for review). For example, at least a part of the antinociceptive action of NSAIDs is due to Na^+^ channel inhibition, leading to nerve AP conduction inhibition (see [176] for review). In support of this idea, an acetic acid-based NSAID diclofenac inhibited Na^+^ channels in rat DRG neurons (IC_50_ = 14 and 97 μM for TTX-sensitive and TTX-resistant Na^+^ currents, respectively [125]) and frog sciatic nerve CAPs (IC_50_ = 0.94 mM [126]). When examined in TRP expressed in heterologous cells, diclofenac activated TRPA1 (at 0.3 mM for channels expressed in *Xenopus laevis* oocytes [127]) while inhibiting TRPM3 (IC_50_ = 18.8 μM for human TRPM3 isoform (TRPM3_1325_) responses activated by a TRPM3 agonist pregnenolone sulfate in HEK293 cells [128]; IC_50_ = 6.2 μM for mouse TRPM3 responses activated by pregnenolone sulfate in HEK293 cells [115]). Another acetic acid-based NSAID etodolac activated TRPA1 in mouse DRG neurons and also TRPA1 (but not TRPV1, TRPV2 and TRPM8) expressed in HEK293 cells [129] while inhibiting frog sciatic nerve CAPs (15% amplitude reduction at 1 mM [126]). With respect to fenamic acid-based NSAIDs, all of tolfenamic acid, meclofenamic acid, mefenamic acid and flufenamic acid inhibited frog sciatic nerve CAPs (IC_50_: 0.29, 0.19 and 0.22 mM, respectively, for tolfenamic acid, meclofenamic acid and flufenamic acid; 16% peak amplitude reduction by mefenamic acid at 0.2 mM [126]). Their NSAIDs inhibited TRPM3 expressed in HEK293 cells (pregnenolone sulfate responses: IC_50_ = 11.1, 13.3, 33.1 and 6.6 μM for tolfenamic acid, meclofenamic acid, flufenamic acid and mefenamic acid, respectively) and TRPC6 expressed in HEK293 cells (TRPC6 agonist hyperforin responses: IC_50_ = 12.3, 37.5, 17.1 and >300 μM for tolfenamic acid, meclofenamic acid, flufenamic acid and mefenamic acid, respectively [130]). Na^+^ channels are reported to be suppressed by tolfenamic acid, mefenamic acid and flufenamic acid (at 100 μM for human Nav1.7 and (TTX-resistant) Nav1.8 expressed in CHO cells [131]; IC_50_ = 0.189 mM for flufenamic acid to inhibit Na^+^-channel currents in rat hippocampal pyramidal neurons [132]). A propionic acid-based NSAID ibuprofen at 1 mM did not affect frog sciatic nerve CAPs [126], but TRPA1 in rat TG neurons was activated by ibuprofen at 10 mM [133]. Such a difference in activity may be due to a difference in the concentration of ibuprofen used. An aryl propionic acid-based NSAID salicylate is reported to inhibit TRPM7 activated by Mg^2+^ depletion in Jurkat T lymphocytes (at 3–30 mM [134]) and Na^+^ channels in rat inferior colliculus neurons (IC_50_ = 1.43 mM [135]). The results of these NSAID actions are summarized in Table 1.

## 9. Other Drugs

Other pain-related drugs that do not fall into the categories mentioned above also act on both Na^+^ and TRP channels. Behavioral studies have demonstrated that a reduced derivative of pyruvic acid, methylglyoxal (MGO; 2-oxopropanal), which is associated with diabetes resulting in neuropathic pain (diabetic peripheral neuropathy), activates both TRPA1 and Nav1.8 in rat primary afferent neurons [136]. Consistent with this observation, rat Nav1.8 expressed in ND7/23 cells was modified by MGO at 0.1 mM [137]. Primary afferent central terminal TRPA1 and TRPV1 in the rat spinal lamina II were activated by MGO (10 mM); this action was due to reactive oxygen species produced by MGO via advanced glycation end-products [138]. Alternatively, it has been reported in rat TG neurons that an agonist for nicotinic acetylcholine-receptors (involved in nociceptive transmission; see [177] for review), nicotine, at 1 mM inhibits Na^+^ channels while enhancing capsaicin-activating TRPV1 responses, both of which effects are not mediated by nicotinic acetylcholine receptors [139]. Although nicotine (0.1 mM) as well as capsaicin increased the spontaneous release of L-glutamate in the rat spinal lamina II, this nicotine action was suggested to be mediated by the activation of presynaptic nicotinic acetylcholine receptors [178]. The possibility cannot be ruled out from the above-mentioned experimental result [139] that primary afferent central terminal TRPV1 activated by nicotine plays a role in the presynaptic facilitation in the lamina II. Moreover, a neuroactive steroid pregnenolone sulfate, which inhibits capsaicin-induced nociception in behavioral studies [140] while being a robust TRPM3 activator (EC_50_ = 23 μM at −80 mV for TRPM3 expressed in HEK293 cells [141]; also see [115]), inhibited Nav1.2 expressed in *Xenopus laevis* oocytes (IC_50_ = 53 μM [142]). Riluzole, which depresses nociceptive responses in mice [179], inhibited TTX-sensitive and TTX-resistant Na^+^ channels in rat DRG neurons (with IC_50_ values of 90 and 143 μM, respectively [143]), and activated TRPC5 expressed in HEK293 cells (EC_50_ = 9.2 μM; [144]).

Anandamide is shown to activate TRPV1 in rat DRG neurons (when intracellularly applied [145]) and TG neurons (when bath-applied; EC_50_ = 10 μM [19]), while suppressing Nav1.2, (TTX-sensitive) Nav1.6, Nav1.7, and Nav1.8 (IC_50_ = 17, 12, 27 and 40 μM, respectively), expressed in *Xenopus laevis* oocytes [146], and rat TTX-sensitive and TTX-resistant Na^+^ channels (IC_50_: 5.4 and 38.4 μM, respectively [147]). Consistent with the fact that anandamide activates not only TRPV1 but also cannabinoid receptors (see [168,180] for review), anandamide inhibited formalin-induced nociception in a fashion sensitive to a CB1 receptor antagonist SR141716A [181]. An inhibition of primary afferent monosynaptically-evoked C-fiber EPSC (involved in slow pain transmission [1]), as produced by capsaicin (1 μM), was not mimicked by anandamide (10 μM) in the rat spinal lamina II, indicating that anandamide does not activate TRPV1 in primary afferent C-fiber central terminals [182]. On the other hand, spontaneous L-glutamate release in rat brain stem lamina II neurons was enhanced by anandamide (by 79% at 30 μM) in a manner sensitive to a TRPV1 antagonist capsazepine [19]. The possibility that primary afferent central terminal TRPV1 is involved in the modulatory effect of anandamide on nociceptive transmission cannot be ruled out. The results of these other drug actions are summarized in Table 1.

## 10. Plant-Derived Compounds

Many plant-derived compounds are involved in modulating analgesic activity when orally, intraperitoneally or intrathecally administrated, with fewer side effects than synthetic chemicals (see [183,184] for review). Among them, representative TRP agonists, capsaicin, AITC and menthol, activated TRPV1 [185], TRPA1 [186,187] and TRPM8 [188,189], respectively, are located in the cell body of primary afferent neuron and TRP-expressing heterologous cells (see [3,4] for review). For example, capsaicin activated TRPV1 with an EC_50_ value of 0.68 μM in rat TG neurons [190,191] and of ca. 0.01 μM in human TRPV1-expressing CHO cells [192]. AITC activated TRPA1 with an EC_50_ value of 22 μM in mouse TRPA1-expressing CHO cells [186], and menthol activated TRPM8 with an EC_50_ value of 80 μM in rat TG neurons [188]. On the other hand, Na^+^ channels were inhibited by capsaicin (IC_50_ = 0.45 μM in capsaicin-sensitive neurons [193]; 85% amplitude reduction of TTX-resistant Na^+^ currents by 1 μM capsaicin [194]) and also by menthol (IC_50_ = 0.571 mM for rat type IIA Na^+^ channels expressed in HEK293 cells [195]).

Similar modulatory actions have been reported for other plant-derived compounds. Anethole (1-methoxy-4-[(*E*)-prop-1-enyl]benzene; a phenolic derivative contained in plants such as anise [196]) activated TRPA1 in mouse DRG and TG neurons, and also in human TRPA1-expressing HEK293 cells (EC_50_ = 80.8 μM [197]), while inhibiting Na^+^ channels in rat DRG neurons (IC_50_ = 1.85 mM [198]). Resveratrol (trans-3,5,4′-trihydroxystilbene; a natural stilbenoid contained in grape [199]) inhibited both mouse TRPA1 expressed in HEK 293 cells (IC_50_ = 0.75 μM for AITC responses [200]) and Na^+^ channels (ca. 25% amplitude reduction at 20 μM for TTX-sensitive Na^+^ currents in rat cortical neurons [201]). Curcumin ((1E,6E)-1,7-bis (4-hydroxy-3-methoxyphenyl)-1,6-heptadiene-3,5-dione; the major component of turmeric [202]), which has two vanillyl groups, suppressed capsaicin-induced responses in TG neurons and TRPV1-expressing HEK293 cells [203], while inhibiting frog sciatic nerve CAPs (10% amplitude reduction at 0.05 mM [204]; see [205] for actions of curcumin on ion channels).

With respect to primary afferent central terminal TRP in the spinal lamina II, plant-derived compounds having an ability to activate the channels inhibited nerve AP conduction, possibly Na^+^ channels, in the frog sciatic nerve. For instance, capsaicin increased spontaneous L-glutamate release by TRPV1 activation (31% sEPSC frequency increase at 1 μM [18]), while suppressing frog sciatic nerve CAPs (36% amplitude reduction at 0.1 mM [204]). Similar spontaneous L-glutamate release enhancement was produced in menthol-induced TRPM8 activation (869% frequency increase at 0.5 mM in postnatal 10–11 days rat [20]; 411% frequency increase at 0.1 mM [25]; 295% frequency increase at 0.5 mM [26]; EC_50_ = 0.263 mM [23]; 409% frequency increase at 0.5 mM [27]) and in AITC-induced TRPA1 activation (202% frequency increase at 0.1 mM [21]; EC_50_ = 0.226 mM [23]; at 0.2 mM [22]; 192% frequency increase at 0.1 mM [24]). On the other hand, menthol and AITC inhibited frog sciatic nerve CAPs (IC_50_: 1.1, 0.93 and 1.5 mM for (-)-menthol, (+)-menthol [206] and AITC [207], respectively).

Moreover, spinal lamina II primary afferent central terminal TRPA1 was activated by various plant-derived compounds, cinnamaldehyde ((2*E*)-3-phenylprop-2-enal, which is contained in cinnamon [46]; 205% sEPSC frequency increase at 0.1 mM [21]; EC_50_ = 0.038 mM [23]; 192% frequency increase at 0.3 mM [24]), eugenol (2-methoxy-4-(2-propenyl) phenol, which is contained in clove and bay leaves [46]; EC_50_ = 3.8 mM [208]), zingerone (4-(4-hydroxy-3-methoxyphenyl)-2-butanone, which is contained in ginger [46]; EC_50_ = 1.3 mM [209]), carvacrol (5-isopropyl-2-methylphenol, which is contained in oregano and thyme essential oils [46]; EC_50_ = 0.69 mM [210]), (+)-carvone ((+)-2-methyl-5-(1-methylethenyl)-2-cyclohexenone, which is contained in caraway [46]; EC_50_ = 0.72 mM [211]), thymol (5-methyl-2-isopropylphenol, which is contained in thyme [46]; EC_50_ = 0.18 mM [212]), 1,8-cineole (1,3,3-trimethyl-2-oxabicyclo[2.2.2]octane, which is contained in eucalyptus and rosemary [213]; EC_50_ = 3.2 mM [27]) and citral (3,7-dimethyl-2,6-octadienal, which is contained in lemongrass [214]; EC_50_ = 0.58 mM [215]). All of the compounds reduced frog sciatic nerve CAP amplitudes (cinnamaldehyde: IC_50_ = 1.2 mM [207]; eugenol and zingerone: IC_50_ = 0.81 and 8.3 mM, respectively [204]; carvacrol: IC_50_ = 0.34 mM [206]; (+)-carvone: IC_50_ = 2.0 mM [206]; thymol: IC_50_ = 0.34 mM [206]; 1,8-cineole: IC_50_ = 5.7 mM [206]; citral: IC_50_ = 0.46 mM [216]).

On the other hand, a spearmint component (-)-carvone and 1,4-cineole (1-methyl-4-(1-methylethyl)-7-oxabicyclo[2.2.1]heptane, which is a minor component in plants containing 1,8-cineole [217]), exhibited both rat spinal lamina II primary afferent central terminal TRPV1, but not TRPA1, activation ((-)-carvone: EC_50_ = 0.70 mM [211]; 1,4-cineole: EC_50_ = 0.42 mM [27]) and frog sciatic nerve CAP inhibition (IC_50_: 1.4 and 7.2 mM, respectively, for (-)-carvone and 1,4-cineole [206]). (±)-Linalool ((±)-3,7-dimethylocta-1,6-diene-3-ol, which is contained in lavender [46]) and its isomer geraniol (trans-3,7-dimethyl-2,6-octadien-1-ol [46]), both of which suppressed frog sciatic nerve CAPs (IC_50_ = 1.7 and 0.53 mM, respectively, for (±)-linalool and geraniol [216]), activated different types of spinal lamina II primary afferent central terminal TRP. Thus, (±)-linalool activated both TRPV1 and TRPA1 (EC_50_ = 2.2 mM), and geraniol activated TRPM8 (EC_50_ = 1.6 mM [218]). An active ingredient of Zanthoxylum piperitum, hydroxy-α-sanshool((2*E*,6*Z*,8*E*,10*E*)-*N*-(2-hydroxy-2-methylpropyl)dodeca-2,6,8,10-tetraenamide), also had an ability to activate both TRPV1 and TRPA1 expressed in HEK cells [219], while inhibiting Na^+^ channels in mouse DRG neurons, Nav1.7 and Nav1.8 [220], and frog sciatic nerve CAPs (ca. 50% amplitude reduction at 0.05 mM [207]). A non-pungent capsaicin-like drug capsiate ((E)-8-methyl-6-nonenoic acid (4-hydroxy-3-methoxyphenyl)methyl ester; isolated from a non-pungent cultivar of CH-19 sweet red pepper; see [221] for review) activated both cloned TRPV1 [222] and TRPA1 (EC_50_ = 2.76 μM [223]), while inhibiting frog sciatic nerve CAPs (22% amplitude reduction at 0.1 mM [204]). Primary afferent central terminal TRPV1 in the spinal lamina II was activated by piperine (1-peperoylpiperidine; a pungent component of black pepper, isolated from the fruits of Piper [46]; EC_50_ = 52.3 μM [224]) that inhibited frog sciatic nerve CAPs (23% amplitude reduction at 0.05 mM; [207]) and Na^+^ channels (IC_50_ = 10 μM for rat TTX-sensitive Nav1.4 [225]). A naturally occurring chalcone, cardamonin (2′,4′-dihydroxy-6′-methoxychalcone; isolated from several plants including Alpinia conchigera [226]), which exhibited analgesia in chronic constriction injury-induced neuropathic pain mice model [227], inhibited human TRPA1 expressed in HEK293 cells, activated by cinnamaldehyde (IC_50_ = 0.454 μM [228]), while suppressing frog sciatic nerve CAPs (ca. 50% amplitude reduction at 3.7 mM [229]). It has been preliminarily shown by molecular docking and molecular dynamic simulation that cardamonin acts on Nav1.7 [230].

CAP inhibition similar to that in the frog sciatic nerve has been reported in mammalian nerve fibers. For instance, in the rat sciatic nerve, eugenol suppressed CAPs (almost completely block at 0.6 mM [231]), and 1,8-cineole did so with an IC_50_ value of 6–8 mM [232]. Citral, carvacrol, citronellol, (-)-carvone and (+)-carvone inhibited rat sciatic nerve CAPs with the IC_50_ values of 0.23 [233], 0.50 [234], 2.2 [235], 10.7 and 8.7 mM [236], respectively.

Consistent with the plant-derived compound-induced CAP inhibition, Na^+^ channels were suppressed by cinnamaldehyde (IC_50_ = 1.5 and 0.8 mM for TTX-sensitive and TTX-resistant Na^+^ currents, respectively, in mouse TG neurons [237]), eugenol (IC_50_ = 0.308 and 0.543 mM for TTX-sensitive and TTX-resistant Na^+^ currents, respectively, in rat DRG neurons [238]), zingerone (IC_50_ = 23.7 μM in pituitary tumor (GH3) cells [239]), carvacrol (IC_50_ = 0.37 mM in rat DRG neurons [234]), (-)-carvone (IC_50_ = 0.75 mM for Nav1.2 [240]), thymol (IC_50_ = 0.149 mM for rat type IIA Na^+^ channels expressed in HEK293 cells [195]), 1,8-cineole (partial blockade at 6 mM in rat superior cervical ganglion neurons [241]) and (±)-linalool (51% current amplitude reduction at 6 mM in rat DRG neurons [242]; see [243] for review).

As seen in capsaicin, AITC, menthol, anethole, resveratrol and curcumin activities, primary afferent (DRG and TG) neuron cell body and cloned TRP were also modulated by various plant-derived compounds, including cinnamaldehyde (EC_50_ = 61 μM for mouse TRPA1 activation in CHO cells [186]), eugenol (TRPV1 activation in HEK293 cells and TG neurons [244]), zingerone (EC_50_ = 15 mM for TRPV1 activation in rat TG neurons [190]), carvacrol (at 0.3 mM for human and rodent TRPA1 activation in HEK293 cells [245]; EC_50_ = 0.49 mM for TRPV3 activation in HEK293 cells [246]), (-)-carvone (EC_50_ = 1.3 mM for TRPV1 activation in HEK293 cells [247]; TRPV3 activation [246]), thymol (EC_50_ = 0.127 mM for human TRPA1 activation in HEK293 cells [248]; EC_50_ = 0.0522 mM for rat TRPM8 activation in HEK293 cells [249]; at 1 mM for human TRPV3 activation in HEK293 cells [250]), piperine (EC_50_ = 0.035 mM for TRPV1 activation in rat TG neurons [190]), citral (EC_50_ = 0.465, 0.926 and 0.0335 mM for rat TRPV1, TRPV3 and TRPM8 activation in HEK293 cells, respectively, and ca. 0.2 mM for rat TRPA1 activation threshold [251]), 1,8-cineole (at 5 mM for human TRPM8 activation in HEK293 cells; IC_50_ = 3.4 mM for AITC response inhibition [252]), 1,4-cineole (at 5 mM for human TRPM8 and TRPA1 activation in HEK293 cells [252]), (±)-linalool (EC_50_ = 6.7 mM for mouse TRPM8 activation in HEK293 cells [253]; EC_50_ = 0.117 mM for rat TRPA1 activation in HEK293 cells [254]) and geraniol (EC_50_ = 5.9 mM for mouse TRPM8 activation in HEK293 cells [253]; at 1 mM for each of rat TRPV1, TRPA1 and TRPM8 activation in CHO cells [255]). The results for these plant-derived compound actions are summarized in Table 2.

Traditional Japanese and Chinese medicines are used for a variety of purposes, including pain relief (see [256,257] for review). Many antinociceptive drugs used in such folk medicines, which are derived from plants, suppress sciatic nerve CAPs [258,259], possibly through Na^+^ channel inhibition. For example, a traditional Japanese medicine daikenchuto, which has an ability to alleviate intestinal obstruction-related abdominal pain [260], inhibited frog sciatic nerve CAPs (IC_50_ = 1.1 mg/mL [258]). Another traditional Japanese medicine inchinkoto, used for oral mucositis (see [261] for review), suppressed frog sciatic nerve CAPs (IC_50_ = 5.4 mg/mL [259]). Mitigation by inchinkoto of oral mucositis-related pain has been attributed to eugenol having an ability to modulate TRP (see [259]). Astragaloside, isolated from a traditional Chinese medicinal plant, which alleviates inflammatory pain [262], inhibited frog sciatic nerve CAPs (25% amplitude reduction at 50 mg/mL [263]). Since many plant-derived compounds modulate TRP, it is possible that traditional medicinal drugs act on both Na^+^ and TRP channels. In support of this idea, daikenchuto is shown to contain TRP agonists, such as hydroxy-α-sanshool (for instance, see [264]). This issue remains to be further examined.

## 11. Voltage-Gated Na^+^ Channels and TRP Channels

As mentioned above, the chemical compounds that affect nerve AP conduction and/or Na^+^ channels also modify TRP activation (see Table 1 and Table 2). Consistent with this result, TRP channels are similar in chemical structure to Na^+^ channels. TRP channels are composed of homo- or hetero-tetramers of subunits having six membrane-spanning segments [4], while the α subunit of the Na^+^ channels, which forms an ion conduction pore, consists of four repeated domains, each of which contains six membrane-spanning segments [265]. In either case, their six segments (S1-S6) contain an ion-permeable pathway formed by S5, S6 and the intervening pore loop region. Their detailed chemical structures have been revealed by using single particle analysis with cryo-electron microscopy (see [266,267,268] for TRPV1; [269] for TRPA1; [270,271] for TRPM8; [272] for TRPM2; [273] for TRPM3; [274] for TRPM7; [275] for TRPC3; [276] for TRPC5; [275,277] for TRPC6; see [278] for review of Na^+^ channels). The compounds given in Table 1 and Table 2 may act on channel proteins themselves or on cell membrane lipids in close proximity to the proteins, and thereby modify ion channel function. Since their channel proteins have intracellular sites that undergo phosphorylation by second messengers to alter channel function [4,265], the compounds may not act directly on those channels. Although the structure–function relationships of Na^+^ and TRP channels have been studied in detail [279,280], it is beyond the scope of this review to describe at which sites on these channels the various compounds presented in this review act. In the future, it will be necessary to clarify a common site in two types of the TRP and Na^+^ channels on which each of the compounds acts by using molecular docking methods, which allow detailed understanding of the interaction between ion channels and ligands (for example, see [281] for review).

When considering the role of Na^+^ and TRP channels in nociceptive transmission, it may be important to be aware of the differences in the efficiency of drug action on these channels. However, because it is unlikely that a drug would act simultaneously on the peripheral and central terminals and nerve fibers of primary afferent neurons, it may be difficult to obtain meaningful concentration-dependence for drug action via comparisons of the efficacy of a drug acting on each. With respect to practical application, considering that the mM order values of many analgesic substances for CAP inhibition are higher than those required to activate TRPV1 or TRPA1 in primary afferent neurons, topically administrated substances are likely to produce analgesia by inhibiting pain conduction rather than by activating TRPV1 or TRPA1 in the peripheral terminals and causing pain.

## 12. Conclusions

Nociceptive pain is a physiological mechanism useful to protect a person against injury to tissues, while neuropathic pain is due to a long-lasting excessive increase in the excitability of neurons near injured neuronal tissues and thus negatively impacts on the quality of life and well-being [1].Clinically, nociceptive pain is alleviated by opioids and NSAIDs, while neuropathic pain is mitigated by using analgesic adjuvants, such as local anesthetics, antiepileptics, antidepressants and α_2_-adrenoceptor agonists. General anesthetics used for consciousness loss in surgery also cause analgesia. Many plant-derived compounds produce analgesia, as evidenced by the use of herbal medicines to relieve pain in traditional medicine. Other drugs, such as methylglyoxal, nicotine, pregnenolone sulfate, riluzole and anandamide, also have an antinociceptive action. This review demonstrated that their chemical compounds exhibit both TRP and Na^+^ channel modulation. This result indicates that a modulatory site on TRP channel protein may be similar in chemical structure to that of Na^+^ channel protein. It remains to be clarified what structures common to TRP and Na^+^ channels the drugs act on, and what drugs beside those mentioned here act on both. Primary afferent peripheral terminal TRP activation produces APs, while Na^+^ channel inhibition results in depressed AP conduction from the periphery to the CNS. Primary afferent central terminal TRP activation in the spinal and brain stem lamina II leads to nociceptive transmission modulation. Clarifying the common site of drug action in TRP and Na^+^ channels may be useful for development of a new analgesic substance that act on both of the channels near the peripheral and central terminals of primary afferent neuron.

## Figures and Tables

**Table 2 biomolecules-14-01619-t002:** Plant-derived compounds exhibiting both TRP and Na^+^ channel (or CAP) modulation.

Plant-Derived Compounds	Central Terminal TRP Modulation	Primary Afferent Neuron and Cloned TRP Modulation	Voltage-Gated Na^+^ Channel	CAP	References
Capsaicin	TRPV1 ↑	TRPV1 ↑	↓	↓	[18,20,185,190,191,192,193,194,204]
AITC	TRPA1 ↑	TRPA1 ↑	n.d.	↓	[21,22,23,24,186,187,207]
Menthol	TRPM8 ↑	TRPM8 ↑	↓	↓	[20,23,25,26,27,188,189,195,206]
Anethole	n.d.	TRPA1 ↑	↓	n.d.	[197,198]
Resveratrol	n.d.	TRPA1 ↓	↓	n.d.	[200,201]
Curcumin	n.d.	TRPV1 ↓	n.d.	↓	[203,204]
Cinnamaldehyde	TRPA1 ↑	TRPA1 ↑	↓	↓	[21,23,24,186,207,237]
Eugenol	TRPA1 ↑	TRPV1 ↑	↓	↓	[204,208,231,238,244]
Zingerone	TRPA1 ↑	TRPV1 ↑	↓	↓	[190,204,209,239]
Carvacrol	TRPA1 ↑	TRPA1 ↑ TRPV3 ↑	↓	↓	[206,210,234,245,246]
(+)-Carvone	TRPA1 ↑	n.d.	n.d.	↓	[206,211,236]
Thymol	TRPA1 ↑	TRPA1 ↑ TRPM8 ↑ TRPV3 ↑	↓	↓	[195,206,212,248,249,250]
1,8-Cineole	TRPA1 ↑	TRPM8 ↑ TRPA1 ↓	↓	↓	[27,206,232,241,252]
Citral	TRPA1 ↑	TRPV1 ↑ TRPV3 ↑ TRPM8 ↑ TRPA1 ↑	n.d.	↓	[215,216,233,251]
(-)-Carvone	TRPV1 ↑	TRPV1 ↑ TRPV3 ↑	↓	↓	[206,211,236,240,246,247]
1,4-Cineole	TRPV1 ↑	TRPM8 ↑ TRPA1 ↑	n.d.	↓	[27,206,252]
(±)-Linalool	TRPV1 ↑ TRPA1 ↑	TRPM8 ↑ TRPA1 ↑	↓	↓	[216,218,242,253,254]
Geraniol	TRPM8 ↑	TRPM8 ↑ TRPV1 ↑ TRPA1 ↑	n.d.	↓	[216,218,253,255]
Hydroxy-α-sanshool	n.d.	TRPV1 ↑ TRPA1 ↑	↓	↓	[207,219,220]
Capsiate	n.d.	TRPV1 ↑ TRPA1 ↑	n.d.	↓	[204,222,223]
Piperine	TRPV1 ↑	TRPV1 ↑	↓	↓	[190,207,224,225]
Cardamonin	n.d.	TRPA1 ↓	n.d.	↓	[228,229,230]

n.d.: not determined; ↑: activation; ↓: inhibition.

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
