# Peer review of "Anesthetic- and Analgesic-Related Drugs Modulating Both Voltage-Gated Na+ and TRP Channels"

_biomolecules, 2024, doi:10.3390/biom14121619_

Round 1

Reviewer 1 Report

Comments and Suggestions for Authors

The manuscript summarizes how anesthetic and analgesic drugs affect pain transmission by modulating voltage-gated Na+ channels and TRP channels, providing a detailed review of how various drugs impact these channels, including local anesthetics, general anesthetics, antiepileptic drugs, and opioids. Overall, the manuscript offers a comprehensive overview of the effects of multiple drugs on pain transmission, particularly in terms of their modulation of Na+ and TRP channels. It is recommended that the authors carefully proofread the article and further discuss the practical applications of their findings to enhance the overall quality of the manuscript.

Several minor concerns are listed:

1, The inhibition of sodium channels mediates analgesic activity, while the activation of TRPV1/TRPA1 leads to pain; it is logically inconsistent for local anesthetics to inhibit voltage-gated sodium channels but emphasize the activation of TRPV1/TRPA1. It is recommended to emphasize the concentration-effect relationships on different channels, which could help explain this query.

2, It is recommended that the manuscript be supplemented with an introduction to the neural circuits involving voltage-gated sodium channels and TRP channels in the context of analgesia and anesthesia.

3, Descriptions of how drugs interact with Na+ and TRP channels may be insufficiently detailed, particularly regarding recent advances in structural biology.

4, The review may lack discussion on the potential clinical impact of these findings, such as new drug development or novel uses for existing medications.

5, A comparative analysis of the mechanisms and effects of different drugs and compounds may be lacking, which would help readers understand their similarities and differences.

Author Response

I would like to sincerely thank you for your valuable comments and suggestions on my manuscript.  I deeply appreciate the time and effort you dedicated to reviewing my manuscript.  All your suggestions and changes have been included in the manuscript as requested and the corrected parts are highlighted in red in the manuscript.  Below are the changes made to each point that was suggested.

The manuscript summarizes how anesthetic and analgesic drugs affect pain transmission by modulating voltage-gated Na+ channels and TRP channels, providing a detailed review of how various drugs impact these channels, including local anesthetics, general anesthetics, antiepileptic drugs, and opioids. Overall, the manuscript offers a comprehensive overview of the effects of multiple drugs on pain transmission, particularly in terms of their modulation of Na+ and TRP channels. It is recommended that the authors carefully proofread the article and further discuss the practical applications of their findings to enhance the overall quality of the manuscript.

My answer: thank you very much for your valuable instruction.  I have proofread my manuscript very carefully.  With respect to the practical application, considering that the mM order values of many analgesic substances for CAP inhibition are higher than those required to activate TRPV1 or TRPA1 in primary afferent neurons, topically administrated substances are likely to produce analgesia by inhibiting pain conduction rather than by activating TRPV1 or TRPA1 in the peripheral terminals and causing pain.  This idea has been mentioned in the first paragraph on page 13.

Several minor concerns are listed:

1, The inhibition of sodium channels mediates analgesic activity, while the activation of TRPV1/TRPA1 leads to pain; it is logically inconsistent for local anesthetics to inhibit voltage-gated sodium channels but emphasize the activation of TRPV1/TRPA1. It is recommended to emphasize the concentration-effect relationships on different channels, which could help explain this query.

    My answer: thank you very much for your valuable suggestion.  It may be important to emphasize this point.  However, since it is unlikely that a drug would act simultaneously on the peripheral and central terminals, and nerve fibers of primary afferent neurons, it may be difficult to obtain meaningful concentration-dependence of drug action from comparisons of the efficacy of a drug acting on each.  Based on this idea, “When considering the role of Na+ and TRP channels in nociceptive transmission, it may be important to be aware of the differences in the efficiency of drug action on these channels.  However, because it is unlikely that a drug would act simultaneously on the peripheral and central terminals, and nerve fibers of primary afferent neurons, it may be difficult to obtain meaningful concentration-dependence of drug action from comparisons of the efficacy of a drug acting on each.” have been given in the fourth paragraph on page 12.

2, It is recommended that the manuscript be supplemented with an introduction to the neural circuits involving voltage-gated sodium channels and TRP channels in the context of analgesia and anesthesia.

   My answer: thank you very much for your valuable suggestion.  According to your instruction, “Regarding nociceptive transmission, in primary afferent neurons, which transmit information as APs involving Na+ channel activation, TRPs at their peripheral terminals receive nociceptive stimuli, while TRPs at their central terminals modify the transmission of nociceptive information.” has been given in the second paragraph on page 2. 

3, Descriptions of how drugs interact with Na+ and TRP channels may be insufficiently detailed, particularly regarding recent advances in structural biology.

   My answer: thank you very much for your valuable instruction.  With respect to this matter, “Although the structure-function relationships of Na+ and TRP channels have been studied in detail [Catterall et al., 2014; Owsianik et al., 2006], it is beyond the scope of this review to describe at which sites on these channels the various compounds presented in this review act.” has been given in the third paragraph on page 12.

Catterall, W.A. Structure and function of voltage-gated sodium channels at atomic resolution. Exp. Physiol. 2014, 99.1, 35–51.

Owsianik, G.; D’hoedt, D.; Voets, T.; Nilius, B. Structure–function relationship of the TRP channel superfamily. Rev. Physiol. Biochem. Pharmacol. 2006, 156, 61-90.

4, The review may lack discussion on the potential clinical impact of these findings, such as new drug development or novel uses for existing medications.

   My answer: thank you very much for your valuable suggestion.  Clarifying the common site of drug action in TRP and Na+ channels may be useful for development of a new analgesic substance that act on both of the channels near the peripheral and central terminals of primary afferent neuron.  Please note that there are various subtypes of voltage-gated Na+ channels and that it is possible that peripheral and central terminal TRP channels are different in property from each other.  In the last of this manuscript (lines 530 and 531 on page 13), “.. that act on both of the channels near the peripheral and central terminals of primary afferent neuron” has been added.

5, A comparative analysis of the mechanisms and effects of different drugs and compounds may be lacking, which would help readers understand their similarities and differences.

   My answer: I am grateful for such comments.  According to your instruction, the mechanism of drug action has been added in sections where it was not previously described.   “by activating opioid receptors” has been given in the third paragraph on page 5, and “by activating α2 adrenoceptors” in the second paragraph on page 6.  With respect to the other drugs, their mechanisms and effects have been already mentioned in the original manuscript.

Reviewer 2 Report

Comments and Suggestions for Authors

The manuscript is a very detailed account of our knowledge about agents that act on both voltage-gated Na channels and TRP channels. The aim of the work is not fully clear to me: what can we do with this gathered information?

Another problem is the organization of the review: the author has made no effort to categorize the agents acting on these targets. A simple classification could be a separation of drugs whose primary molecular target is the voltage-gated channels (e.g., local anesthetics, certain antiepileptics) and those that have another primary target (e.g. opioids or NSAIDs).

In several studies, very high IC50 values were obtained (even in the millimolar range) raising concerns about the in vivo relevance of the results. I would welcome any general conclusion (if possible) on the comparison of IC50 values for Na channels and TRP channels if the direction of modulation is the same (e.g. inhibition).

Throughout the text: the correct spelling is "primary afferent neuron" and not "primary-afferent neuron".

 Throughout the text: if a TRP channel activator is applied to the central terminals of nociceptive primary afferent neurons and glutamate release is increased, it is considered evoked release and not spontaneous one.

 Line 10: what is meant by "nociceptive transmission in the peripheral terminals of primary afferent neurons"?

 Regarding the inhibitory effects of opioids on the voltage-gated Na channels can it be a secondary action due to opioid-induced K channel opening leading to membrane hyperpolarization?

Author Response

I would like to sincerely thank you for your valuable comments and suggestions on my manuscript.  I deeply appreciate the time and effort you dedicated to reviewing my manuscript.  All your suggestions and changes have been included in the manuscript as requested and the corrected parts are highlighted in red in the manuscript.  Below are the changes made to each point that was suggested.

The manuscript is a very detailed account of our knowledge about agents that act on both voltage-gated Na channels and TRP channels. The aim of the work is not fully clear to me: what can we do with this gathered information?

My answer: the aim of this study was to summarize the effects of local anesthetics, general anesthetics, antiepileptics, opioids, adrenoceptor agonists, antidepressants, NSAIDs and plant-derived compounds on both TRP and Na+ channels (nerve AP conduction) and further to know whether the other pain-related drugs exhibit a similar effect.  With respect to this matter, “and the other pain-related drugs” has been added in line 89 on page 2.  By doing so, it may be possible to develop new pain-relieving substances, as described at the end of page 13.

Another problem is the organization of the review: the author has made no effort to categorize the agents acting on these targets. A simple classification could be a separation of drugs whose primary molecular target is the voltage-gated channels (e.g., local anesthetics, certain antiepileptics) and those that have another primary target (e.g. opioids or NSAIDs).

My answer: thank you very much for your thoughtful comments.  As pointed out by you, there may be a way to classify and describe them in terms of voltage-gated channel and so on, but since there are many reports showing that opioids and NSAIDs act on voltage-gated channels, I have decided to classify and describe them in terms of analgesic adjuvants (local anesthetics, antiepileptics ..) and analgesics (opioids and NSAIDs).  To demonstrate this point, the first sentence of the sections on local anesthetics, antiepileptics, α2-adrenoceptor agonist and antidepressants have already stated that they are analgesic adjuvants.  In order to make my idea more clear, “local anesthetics, antiepileptics, α2-adrenoceptor agonists, antidepressants (all of which are used as analgesic adjuvants),” has been mentioned in lines 17 and 18 in Abstract.   

In several studies, very high IC50 values were obtained (even in the millimolar range) raising concerns about the in vivo relevance of the results. I would welcome any general conclusion (if possible) on the comparison of IC50 values for Na channels and TRP channels if the direction of modulation is the same (e.g. inhibition).

My answer: thank you very much for your valuable comments.  Although the concentrations of some drugs that affect CAPs and Na+ channels are high, this is the concentration likely to be encountered when the drugs are given topically around neural tissue.  Based on this idea, “With respect to the practical application, considering that the mM order values of many analgesic substances for CAP inhibition are higher than those required to activate TRPV1 or TRPA1 in primary afferent neurons, topically administrated substances are likely to produce analgesia by inhibiting pain conduction rather than by activating TRPV1 or TRPA1 in the peripheral terminals and causing pain.” have been given in the first paragraph on page 13.

    With respect to the comparison of IC50 values for Na channels and TRP channels, since it is unlikely that a drug would act simultaneously on the peripheral and central terminals, and nerve fibers of primary afferent neurons, it may be difficult to obtain meaningful concentration-dependence of drug action from comparisons of the efficacy of a drug acting on each.  Based on this idea, “When considering the role of Na+ and TRP channels in nociceptive transmission, it may be important to be aware of the differences in the efficiency of drug action on these channels.  However, since it is unlikely that a drug would act simultaneously on the peripheral and central terminals, and nerve fibers of primary afferent neurons, it may be difficult to obtain meaningful concentration-dependence of drug action from comparisons of the efficacy of a drug acting on each.” have been given in the fourth paragraph on page 12.

Throughout the text: the correct spelling is "primary afferent neuron" and not "primary-afferent neuron".

My answer: as instructed by you, "primary-afferent neuron" has been changed to "primary afferent neuron" throughout this manuscript.

 Throughout the text: if a TRP channel activator is applied to the central terminals of nociceptive primary afferent neurons and glutamate release is increased, it is considered evoked release and not spontaneous one.

My answer: thank you very much for your thoughtful comment.  When a TRP channel activator acts on the central terminals of nociceptive primary afferent neuron, TRP channel opens, resulting in a non-selective influx of cations.  The resulting membrane depolarization activates voltage-gated Ca2+ channels, and Ca2+ passes through the TRP channel, increasing intracellular Ca2+ concentration, which then causes the enhanced spontaneous release of L-glutamate.  “Evoked release” seems to be generally used for neurotransmitter release as a result of production of action potentials in nerve terminals.  The above-mentioned idea has been added in the first paragraph on page 2, as “The enhanced spontaneous L-glutamate release is caused by an increase in intracellular Ca2+ concentration, which is due to activation of voltage-gated Ca2+ channels via membrane depolarization occurring as a result of opening of TRP, or by Ca2+ passing through the TRP itself [3].”    

Line 10: what is meant by "nociceptive transmission in the peripheral terminals of primary afferent neurons"?

My answer: thank you very much for such a comment.  “mediate nociceptive transmission” has been changed to “are involved in receiving and transmitting nociceptive stimuli” in line 10 of Abstract. 

 Regarding the inhibitory effects of opioids on the voltage-gated Na channels can it be a secondary action due to opioid-induced K channel opening leading to membrane hyperpolarization?

My answer: thank you very much for your idea about an involvement of opioid-induced K+ channel opening in the inhibitory effects of opioids on the voltage-gated Na+ channels.  It is unlikely that opioid-induced K+ channel opening (hyperpolarization) is involved in voltage-gated Na+ channel current reduction produced by opioid under the voltage-clamp condition.  With respect to frog sciatic nerve CAP inhibitions produced by morphine and tramadol, these were resistant to a non-selective opioid-receptor antagonist naloxone and thus were not mediated by opioid receptors whose activation leads to membrane hyperpolarization.  Such an idea has been stated in the first paragraph on page 6 as “The frog sciatic nerve CAP inhibitions produced by morphine and tramadol may be due to a membrane hyperpolarization occurring as a result of opioid-receptor activation.  However, this is unlikely, because the morphine and tramadol activities were resistant to a non-selective opioid-receptor antagonist naloxone [64,106].”       

Round 2

Reviewer 2 Report

Comments and Suggestions for Authors

The authors have adequately addressed all my comments and concerns. I have only one suggestion to change the title of the paper to the one below:

Anesthetic- and Analgesic-Related Drugs  Modulating Both Voltage-Gated Na+ and TRP Channels

Author Response

Thank you very much for reviewing the revised manuscript.  As requested by you, I have corrected the title.